# Boosting Whole-Grain Utilization in the Consumer Market: A Case Study of the Oldways Whole Grains Council’s Stamped Product Database

**DOI:** 10.3390/nu14030713

**Published:** 2022-02-08

**Authors:** Caroline Sluyter, Kelly LeBlanc, Kristen Hicks-Roof

**Affiliations:** 1Oldways Whole Grains Council, Boston, MA 02116, USA; kelly@oldwayspt.org; 2Department of Nutrition and Dietetics, University of North Florida, Jacksonville, FL 32224, USA; hicks.roof@unf.edu

**Keywords:** whole-grain content, Latin America, whole-grain trends, food labeling, international

## Abstract

Whole grains are a vital part of a healthy diet, yet there are insufficient data on the whole-grain content of commercial food products. The purpose of this research is to examine the long-term change in (1) measured whole grains in food products, (2) Whole Grain Stamp usage and (3) the prominence of whole-grain ingredients and product categories, across the United States and Latin America. These changes were quantified by analyzing the Oldways Whole Grains Council’s (WGC) Stamped Product Database from 2007 to 2020. Mean whole grains increased 36–76%, from 19 grams to 25.8 grams per serving in the U.S. and 18.1 grams to 31.9 grams per serving in Latin America. Whole Grain Stamp usage worldwide has increased from 250 products in 2005 to more than 13,000 products in 2020. These findings suggest that manufacturers are increasing the percentage of the grain that is whole in their products and developing more whole-grain products for consumers, thus providing an opportunity for consumers to meet national-level whole-grain recommendations.

## 1. Introduction

Whole grains are widely recognized to be an important part of a healthy diet by both U.S. [1] and global [2] dietary guidelines. The Dietary Guidelines for Americans recommends making at least half of all grains whole, thereby consuming at least 48 grams of whole grain per day [1]. Dietary guidelines in Latin America tend to be more qualitative than quantitative [3], and many Latin American countries do not include an explicit whole-grain recommendation [4]. NHANES data indicate that U.S. whole-grain consumption has been slowly but steadily increasing, reaching 14.4 grams of whole-grain per day (0.9 ounce-equivalents) in 2013–2014 [5]. Similarly, the average person in Latin America eats only 14.7 grams of whole-grain foods per day [6].

Whole grains are an important component of a healthy, balanced diet. Whole-grain intake is consistently associated with lower rates of colorectal cancer [7,8], type 2 diabetes [9], cardiovascular disease, and mortality [10], as well as beneficial changes to the gut microbiome [11]. On the other hand, suboptimal whole-grain intake is associated with serious health risks. The Global Burden of Disease study found that poor diet played a role in about 20% of all deaths globally and that low whole-grain intake was the second greatest dietary risk factor for mortality after high sodium consumption [12]. Therefore, it is clear there is much to be gained by increasing whole-grain intake globally. 

In 2005, the Oldways Whole Grains Council introduced the Whole Grain Stamp packaging symbol [13], part of a voluntary third-party certification program. The Stamp helps consumers quantify whole-grain intake by communicating how many grams of whole grain are in one serving of a product, thereby enabling consumers to increase intake more easily toward recommended levels [14]. Today, the Whole Grain Stamp program has three different types of Stamps: the 100% Stamp (which indicates that 100% of the grain ingredients are whole grain), the 50%+ Stamp (which indicates that at least 50% of the grain ingredients are whole grain) and the Basic Stamp (which indicates that a product contains at least 8 grams of whole grain per serving but may have more refined grain than whole grain) (Figure 1). 

While other whole-grain packaging symbols exist in Australia [15], Denmark [16] and Singapore [17], the Oldways Whole Grains Council’s WGC Stamped Product Database is the largest database of whole-grain products that includes the gram amount of whole grain in the product as well as the product type (bread, snack, etc.) and types of whole grains (whole wheat, oats, etc.) incorporated into the product.

The lack of published data on whole-grain content of foods within the U.S. and Latin America limits the ability of researchers to explore whether or not availability of whole-grain products may be a factor contributing to low whole-grain consumption. The purpose of this research is to examine the long-term change in (1) measured whole grains in food products, (2) Whole Grain Stamp usage and (3) the prominence of whole-grain ingredients and product categories across the United States and Latin America.

## 2. Materials and Methods

The Whole Grain Stamp is tightly regulated by the Oldways Whole Grains Council, to serve as a symbol for consumers to recognize and understand whole-grain composition in a food product. When a manufacturer applies for use of the Whole Grain Stamp, they submit information about the ingredients, nutritional profile, and whole-grain content of the product to the Oldways Whole Grains Council for review. Products are resubmitted for review every time there is a change in ingredients, formulation, nutrition facts, serving size, or UPC code. Approved products are included in the WGC Stamped Product Database, allowing for the observation of trends over time. As the WGC Stamped Product Database contains proprietary information from food manufacturers, some data cannot be published. However, product name, Stamp type, and total grams of whole grains are publicly available information. The WGC Stamped Product Database has historically recorded data each year from 2007 (when the WGC Stamped Product Database transitioned from paper records to online records) to 2020.

### 2.1. Measured Whole Grains in Food Products

The quantity of whole grain was calculated by taking the gram amount of whole grain displayed on each of the products with the approved Whole Grain Stamp and averaging the products’ whole-grain content on an annual basis. The average whole-grain content was calculated for both U.S. products (2008–2020), and Latin American products (2009–2020) separately.

### 2.2. Whole Grain Stamp Utilization

The growth in Whole Grain Stamp use was measured by the growth in the cumulative number of products with an approved Stamp from 2005 to 2020. The products approved for Stamp use were categorized as follows: (1) those approved for Stamp use in the U.S. only, (2) those approved for Stamp use in the U.S. and at least one international country, (3) those approved for Stamp use in one or more international countries, but not the U.S., and (4) those approved for Stamp use in Latin America (products in this category are not exclusive of those in categories 2 and 3).

### 2.3. Whole-Grain Ingredients

In order to track changes in the prevalence of specific ‘ancient’ whole-grain ingredients, ingredient lists were searched in each year from 2010 (when a new searchable text box field was first added to capture ingredient lists in the WGC Stamped Product Database) to 2020 for the following terms: ‘amaranth,’ ‘buckwheat,’ ‘farro,’ ‘millet,’ ‘quinoa,’ ‘spelt,’ ‘sorghum,’ and ‘teff.’ While other ‘ancient grain’ varieties exist (e.g., einkorn, Job’s tears, and purple barley) there were insufficient examples of products using these ingredients within the WGC Stamped Product Database. Only ingredients in the English language were captured. Based on the number of products containing each of these ingredients, the percentage of products utilizing each grain was calculated. Products captured in these searches and included in the percentage calculations represent a subset of the overall database (between 29–57% of product registrations utilized the text box ingredients list submission option between 2010 and 2020).

The change in the prominence of sprouted-grain ingredients in the WGC Stamped Product Database (including both U.S. and international products) was calculated by adding up the total number of products marked as containing sprouted grains in each year from 2013 (when a new ‘Contains Sprouted Grains’ field was first added to the WGC Stamped Product Database) to 2020 and by calculating the overall percentage of products utilizing this field in each year. Although there is no regulated definition of what constitutes a sprouted grain, it is generally accepted that a sprouted grain is grain which has been allowed to germinate under controlled conditions, thus transforming the long-term-storage starch of the grain’s endosperm into simpler molecules that are easily digested by the growing plant embryo, making many of the nutrients stored in the grain more bioavailable for human consumption [18].

### 2.4. Whole-Grain Food Product Categories

The change in the prominence of particular product categories over time was calculated by adding up the total number of products (across all U.S. and international products) marked as belonging to 17 product categories for each year and then calculating the percentage of all products captured by each of these categories. The 17 product categories were: baking mixes; bars; beverages; breads; cold cereals; entrees; flours; grain side dishes; hot cereals; pasta; pizza and pizza crust; snacks and crackers; soups; tortillas, wraps, and flatbreads; treats; waffles, pancakes, and French toast; and yogurts.

## 3. Results

### 3.1. Measured Whole Grains in Food Products

Globally, the average whole-grain gram amount shown on the Whole Grain Stamp increased from 19 grams to 25.8 grams per serving (a 36% increase) from 2008 to 2020. For products registered for Whole Grain Stamp use in the U.S., a similar increase from 19 grams to 25.1 grams was observed from 2008 to 2020 (Figure 2). In contrast, the growth in Latin America was greater, with an increase from 18.1 grams to 31.9 grams per serving (a 76% increase) from 2009 to 2020.

### 3.2. Whole Grain Stamp Utilization

In the U.S., use of the Whole Grain Stamp has increased significantly (from 250 products in 2005 to more than 10,700 products in 2020) (Figure 3). Since 2009, the proportion of products available outside the U.S. has also increased. As of 2020, more than 27% of Stamped products were available outside the U.S. Stamp use on products in Latin America has also grown steadily, with more than 500 new whole-grain products registered for the Whole Grain Stamp between 2017–2020 (Figure 4). In 2020, over 40% of all products registered for Stamp use outside the U.S. were registered in Latin America.

### 3.3. Whole-Grain Ingredients

Of the eight ‘ancient grains’ included in the analysis, only five (amaranth, millet, quinoa, sorghum, and teff) showed steady or positive change in relative prevalence between 2010 and 2020. Of these five, millet and quinoa are the most common ingredients in products registered for Stamp use. Millet’s prominence has been relatively steady, with inclusion in 8–9% of products. Quinoa was used in under 3% of Stamped products in 2010, but by 2020, it showed up in 10.5% of products—a 3.5-fold increase. In 2010, sorghum was included in 0.6% of products but had increased to 4.4% of Stamped products by 2019 (a 7.4-fold increase). Over 10 years, amaranth’s use doubled, and teff’s prominence quadrupled.

A steady increase in the number of products containing sprouted grains occurred from 2013, when there were just over 100 sprouted products registered for the Stamp, to 2017, when the number of sprouted products grew to nearly 400. Growth since 2017 has leveled off somewhat, but there were more than 400 sprouted products registered for Stamp use in 2020.

### 3.4. Whole-Grain Food Product Categories

The six categories that included the most Stamped products were cold cereals (19.3% of products in 2020); breads (16.7% of products in 2020); snacks and crackers (14.9% of products in 2020); hot cereals (8.2% of products in 2020); grain side dishes (8.1% of products in 2020); and flours (5.8% of products in 2020). Product categories with very few Stamped products included baking mixes; pizza and pizza crusts; waffles, pancakes and French toast; soups; beverages; and yogurts. Among the six most prevalent categories, breads and hot cereals saw declines of 39% and 21%, respectively between 2009 and 2020, while the most significant increase was observed with cold cereal products, which rose 60% in prevalence within the database.

## 4. Discussion

These data indicate that there is has been a stark increase in whole-grain makeup of Stamped products, as well as an increase in products that are registered to use the Whole Grain Stamp. This suggests that manufacturers are prioritizing whole grain in food products, hoping to reduce whole-grain deficits among consumers.

A greater utilization of ‘ancient grains’ such as sorghum and quinoa and increased prominence of sprouted grains supports existing literature on increased consumer interest in ancient and alternative grains [19]. These ancient grains have shown upward trends in food products such as breads, breakfast cereals, baked goods and even beverages [20]. The finding that whole grains are particularly underutilized in the pizza and baking mix categories compared with breads, cereals, snacks, and flour supports existing literature on sources of whole grains in the American diet [21]. These data help to expose product categories where further innovation and development of whole-grain integration may be warranted by food manufacturers. 

Incremental change in the nutritional quality of foods, thus boosting whole-grain consumption, can be achieved by simply increasing the whole-grain content of the food [22]. Given the relatively low consumption of whole grains in the U.S. [5] and Latin America [6], a mean 36% increase in whole-grain content in Stamped products can have profound impacts on whole-grain consumption across the regions. Strategic food labeling, such as the Whole Grain Stamp (Figure 1), which indicates varying levels of whole-grain content, can help meet consumer demand whilst boosting the nutritional profile of commonly consumed foods (e.g., breads, cereals, baked goods). 

With any research study, there are always both strengths and limitations that should be addressed. A major strength of this study is that, to the authors’ knowledge, the WGC Stamped Product Database is the largest database of whole-grain products that includes quantity, product type, and types of grains used in the product. Secondly, this is the first study to describe and compare whole-grain utilization in consumer-level food products across two regions. A limitation of this study is that this analysis does not capture every product on the market, due to the Whole Grain Stamp program being voluntary for manufacturers. This limitation is particularly evident outside the Americas, as Whole Grain Stamp use is most prevalent throughout the U.S., Canada, and Latin America. Another limitation is that only a subset of the products analyzed (29–57%, depending on the year) were searchable for specific whole-grain ingredients (such as sorghum), meaning that trends in whole-grain ingredient usage may not be generalizable to the entire dataset, particularly for non-English language products.

Nutrition programs and policies endorsing whole-grain consumption can only be successful if whole-grain products are readily available in the consumer marketplace. This study demonstrates that food manufacturers are continuing to increase whole-grain content and diverse utilization of food products. Therefore, policy makers, nonprofits, and food manufacturers need to collaborate and communicate to better align whole-grain recommendations with product offerings. More research is needed to better understand whole-grain diversity, utilization, and availability among food products within the Americas and worldwide.

## Figures and Tables

**Figure 1 nutrients-14-00713-f001:**
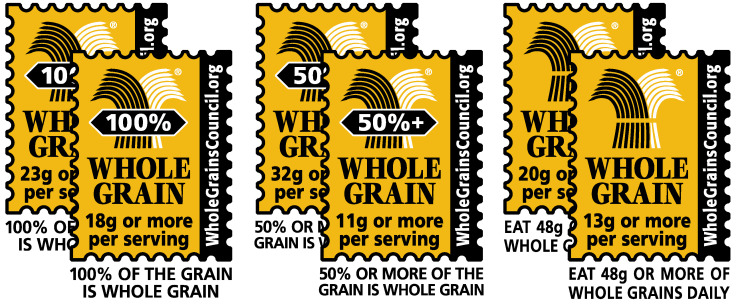
Whole Grain Stamps.

**Figure 2 nutrients-14-00713-f002:**
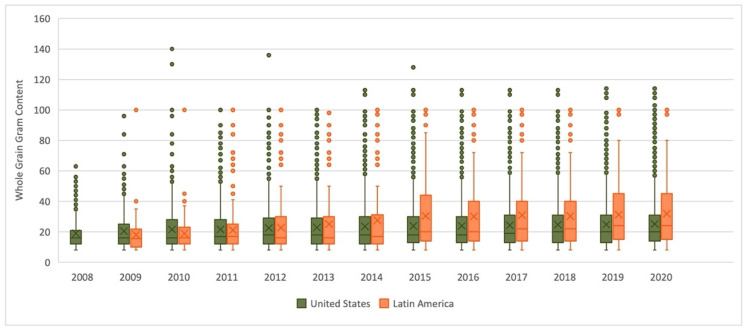
Whole-grain gram content of Whole Grain Stamped products.

**Figure 3 nutrients-14-00713-f003:**
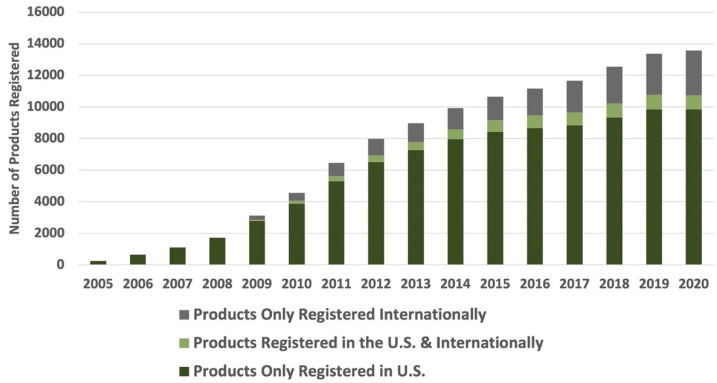
Whole Grain Stamp growth in the U.S. and internationally.

**Figure 4 nutrients-14-00713-f004:**
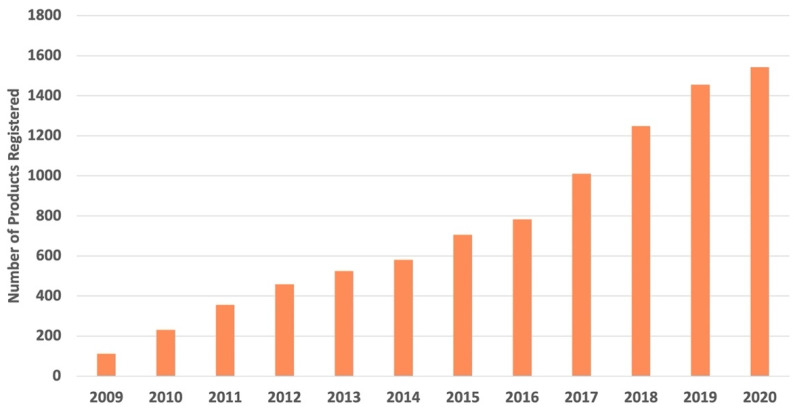
Whole Grain Stamp growth in Latin America.

## Data Availability

Please reach out to the corresponding author for access to the full data set.

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
