# Peer review of "Boosting Whole-Grain Utilization in the Consumer Market: A Case Study of the Oldways Whole Grains Council’s Stamped Product Database"

_nutrients, 2022, doi:10.3390/nu14030713_

Round 1
Reviewer 1 Report
The manuscript “Boosting Whole Grain Utilization in the Consumer Market: A Case Study of the Oldways Whole Grains Council’s Stamped Product Database” by Caroline Sluyter et al. describes the changes in the whole grain utilization within the U.S. and Latin America from 2007 to 2020. The paper is interesting but, in my opinion, the introduction should point out the importance of the presence of whole grain in the diet.
In addition, all the figures may be in higher definition.
Author Response
We would like to thank you for your time in reviewing our manuscript. We feel the revisions were insightful and believe this revised version has been made stronger by your feedback.
Point 1: The paper is interesting but, in my opinion, the introduction should point out the importance of the presence of whole grain in the diet.
Response 1: We fully agree and have added a new second paragraph to the Introduction (with appropriate references included). "Whole grains are an important component of a healthy, balanced diet. Whole grain intake is consistently associated with lower rates of colorectal cancer [7,8], type 2 diabetes [9], cardiovascular disease, and mortality [10], as well as beneficial changes to the gut microbiome [11]. On the other hand, suboptimal whole grain intake is associated with serious health risks. The Global Burden of Disease study found that poor diet played a role in about 20% of all deaths globally and that low whole grain intake was the second greatest dietary risk factor for mortality after high sodium consumption [12]. Therefore, it is clear there is much to be gained by increasing whole grain intake globally."
Point 2: In addition, all the figures may be in higher definition.
Response 2: We have included larger image files for the figures.
Reviewer 2 Report
This is a very interesting article in which authors try to examine the long-term change in measured whole grains in food products, Whole Grain Stamp usage and the prominence of whole grain ingredients and product categories, across the US and Latin America. The paper is well structured and diplays insight into the situation. It adressesan importat issue, not only for dieticians and physicians, but also for the fields of primary and secondary prevention.
However, it needs some minor revisions:
1) The figures are way too small. The information cannot be seen.
Author Response
We would like to thank you for your time in reviewing our manuscript. Your revisions were helpful and we believe this revised version has been made stronger by your feedback.
Point 1: The figures are way too small. The information cannot be seen.
Response 1: We have included larger image files for the figures.
Reviewer 3 Report
Overall, interesting article and I recommend publishing.
Correct the minor typo in the abstract (last line) "providing an opportunity to for consumers"......it seems that "to" should be removed from the sentence.
Line 182...Strategic food labeling is mentioned but no examples of labeling techniques or how this labeling might look are given. This would be helpful to the reader
Author Response
We would like to thank you for your time in reviewing our manuscript. We feel the revisions were insightful and believe this revised version has been made stronger by your feedback.
Point 1: Correct the minor typo in the abstract (last line) "providing an opportunity to for consumers"......it seems that "to" should be removed from the sentence.
Response 1: We have removed "to" from this sentence.
Point 2: Line 182...Strategic food labeling is mentioned but no examples of labeling techniques or how this labeling might look are given. This would be helpful to the reader
Response 2: We agree it was not clear that we were referring to labels like the Whole Grain Stamp. We have edited this sentence as follows: "Strategic food labeling, like the Whole Grain Stamp [Figure 1], which indicates varying levels of whole grain content, can help meet consumer demand whilst boosting the nutritional profile of commonly consumed foods (e.g., breads, cereals, baked goods)."